# Optimizing Precision Probiotics for Mitigating Graft-Versus-Host Disease

**DOI:** 10.3390/microorganisms13040706

**Published:** 2025-03-21

**Authors:** Nonyelum Ebigbo, Apple Long, Phinga Do, Laura Coughlin, Nicole Poulides, Talia Jewell, Shuheng Gan, Xiaowei Zhan, Andrew Y. Koh

**Affiliations:** 1Department of Pediatrics, University of Texas Southwestern Medical Center, Dallas, TX 75390, USA; nonyelum.ebigbo@utsouthwestern.edu (N.E.);; 2Department of Internal Medicine, University of Texas Southwestern Medical Center, Dallas, TX 75390, USA; 3Isolation Bio Inc., San Francisco, CA 94306, USA; 4Department of Population and Data Sciences, University of Texas Southwestern Medical Center, Dallas, TX 75390, USA; 5Harold C. Simmons Comprehensive Cancer Center, University of Texas Southwestern Medical Center, Dallas, TX 75390, USA; 6Department of Microbiology, University of Texas Southwestern Medical Center, Dallas, TX 75390, USA

**Keywords:** probiotics, precision probiotics, short-chain fatty acids

## Abstract

Precision probiotics have shown great promise as novel therapies but have not been fully realized. One major obstacle is that different strains of the same gut microbiota species can induce markedly variable phenotypic outcomes. Here, we aimed to optimize and validate in a preclinical model, a six-species precision probiotic therapy for graft-versus-host disease (GVHD), an autoimmune complication following allogeneic stem cell transplantation. We had identified these six species as associated with protection against GVHD in a prior clinical study. We isolated strains of three of the targeted taxa (*B. longum*, *C. bolteae*, and *Blautia* spp.) from human stem cell transplant patients and characterized their SCFA production in vitro. We observed significant strain-to-strain variability among these gut microbiota taxa in their capacity to produce short-chain fatty acids, a microbiota-derived metabolite shown to be important for mitigating gut GVHD and inflammatory bowel disease, in vitro. We found that *B. longum* was able to augment butyrate production by *C. bolteae* and *Blautia* when co-cultured in vitro. “Optimized” precision probiotics mitigated GVHD and significantly increased survival (*p* = 0.013, log-rank test) in mice compared to a “standard” probiotic consortium of the same bacterial species obtained from a commercial repository. Importantly, the optimized probiotics resulted in significant increases in intestinal short-chain fatty acid concentrations compared to standard probiotics (*p* < 0.001, Mann–Whitney test). Our findings highlight the promising potential of utilizing an optimized precision probiotic approach to maximize therapeutic efficacy.

## 1. Introduction

The gut microbiome, the trillions of microorganisms residing within the human gastrointestinal tract, plays a pivotal role in human health. There is growing evidence that modulating microbiota populations using microbiome-based therapies may serve as an effective therapeutic strategy. Emerging therapies include nutrition modification, delivery of microbial species (via oral probiotics or fecal microbiota transplantation), administration of microbiota-derived metabolites, and/or specific targeting of microorganisms and their associated pathways [1,2]. 

Probiotics, live microorganisms that confer health benefits to the host, have gained substantial traction over the past two decades [3]. The effectiveness of probiotics, however, is often inconsistent, and their therapeutic effect varies widely, even among responders. Recent insights suggest that the efficacy of probiotics is highly strain dependent (i.e., different strains of the same taxa inducing variable phenotypes) [4]. The ability of probiotic strains to colonize the gut, whether temporarily or persistently, differs across individuals and is influenced in part by their baseline resident microbiome, diet, and age. This variability likely contributes to the disparate responses observed among individuals with the same condition receiving identical probiotic formulations [4]. A precision approach to probiotics offers the promise of overcoming these challenges by addressing differences attributed to gut microbiota strain-to-strain variability, host characteristics, and baseline microbiota profiles. 

Graft-versus-host disease (GVHD) is a life-threatening complication of allogeneic hematopoietic stem cell transplantation, characterized by donor T cell-mediated immune responses against recipient tissues, primarily affecting the gastrointestinal tract, skin, and liver. Emerging evidence indicates that gut microbiota dysbiosis plays a pivotal role in GVHD pathogenesis by disrupting intestinal homeostasis, promoting epithelial injury, and exacerbating systemic inflammation [5]. Restoring microbial balance through microbiome-targeted interventions, such as precision probiotics, represents a promising therapeutic strategy to mitigate GVHD severity and improve clinical outcomes. Our previously published study showed that specific commensal bacterial species were noted to be significantly depleted (up to 4 log fold) in pediatric stem cell transplant patients with graft-versus-host disease (GVHD): *Clostridium bolteae*, *Blautia* spp., *Bifidobacterium* spp., *Lactococcus lactis*, *Ruminococcus gnavus*, and *Ruminococcus torques* [5]. Here, we utilized complementary in vitro and in vivo approaches to demonstrate that optimizing precision probiotics can lead to increased therapeutic efficacy for GVHD.

The objective of this study is to evaluate the efficacy of an optimized precision probiotic formulation in mitigating graft-versus-host disease (GVHD) by leveraging strain-specific functional differences in gut microbiota. Specifically, we aimed to isolate and characterize strains from pediatric stem cell transplant patients who did not develop GVHD, assessing their capacity to produce short-chain fatty acids (SCFAs). Furthermore, we seek to assess their therapeutic potential in a preclinical murine model of GVHD. By comparing the optimized precision probiotics to commercial standard probiotic strains, we aim to establish the role of strain-specific metabolic activity in improving clinical outcomes and survival. Ultimately, this study provides foundational evidence for the development of targeted microbiome-based therapies in GVHD and other immune-mediated disorders.

## 2. Materials and Methods

### 2.1. Isolation of Gut Microbiota

Fresh fecal samples were collected from pediatric stem cell transplant patients who did not develop GVHD and stored at −80 °C. All subjects underwent hematopoietic stem cell transplantation at Children’s Medical Center Dallas and were enrolled under the human study protocol (STU 112010-110) through the institutional review boards of the University of Texas Southwestern Medical Center and Children’s Medical Center Dallas.

We utilized two different methods to isolate the gut bacteria of interest from fecal specimens. First, an archived stool aliquot was thawed, diluted with sterile PBS, and inoculated in anaerobic Becton Dickinson (Franklin Lakes, NJ, USA) BACTEC culture media bottles (BD, 442024) supplemented with sterile rumen fluid and incubated at 37 °C under anaerobic conditions. After 24 h of incubation, 100 uL of culture was serially diluted and plated on Yeast extract, Casitone and Fatty acid (YCFA) medium, Reinforced Clostridia (CM0151) medium (ThermoFisher, Waltham, MA, USA, CM0151B), Bacteroides Bile Esculin (BBE) medium (Sigma, St. Louis, MO, USA, B1176), Bifidus Selective Medium (BSM) (Sigma, 88517), and/or Brain heart Infusion/blood (BHI/blood) medium (BD, 237500) and incubated anaerobically at 37 °C for 2–5 days. Individual bacterial colonies were selected and re-streaked onto a fresh plate of the same agar and regrown for 2–5 days. 

For a second complementary method, we utilized the Prospector (Isolationbio, San Carlos, CA, USA), an automated nanoscale array-based platform for bacterial isolation and cultivation which self-sorts individual microbes into 6000+ nanoscale cultivation chambers, thereby allowing the growth of thousands of micro-colonies in parallel. A stool suspension was prepared, loaded unto the culture array, and incubated anaerobically at 37 °C. After 2 days of incubation, the array was imaged under green fluorescence to view the nanoplates with clonal populations and the isolates/colonies were recovered. Isolates were then grown on solid agar media as above. 

To ascertain bacterial species identification, we used a MALDI-TOF (Bruker MALDI Biotyper, Billerica, MA, USA) and/or full length (V1–V9) 16S rRNA gene sequencing. For the latter, genomic DNA was extracted from the bacterial colonies. The 16S rRNA gene was amplified by PCR using universal primers 27F and 1492R [6]. Sanger sequencing was performed, and the NCI Blast database used to assign taxonomic identity to each strain. 

For the standard probiotics, gut microbiota strains were sourced from the American Type Culture Collection (ATCC): *C. bolteae* (human, ATCC, Manassas, VA, USA, BAA-613, strain WAL 16351), *Blautia producta* (human, ATCC 27340, strain VPI 4299), and *Bifidobacterium longum* (human, ATCC 15707, strain E194b). These strains were grown at 37 °C anaerobically per ATCC instructions.

### 2.2. Gut Microbiota Co-Culture Studies

*B. producta*, *C. bolteae*, and *B. longum* were grown individually in YCFA broth media (Anaerobe Systems, Morgan Hill, CA, USA, AS-6805) at 37 °C anaerobically. Gut microbiota were then grown alone or in combination (*B. producta* + *B. longum* and *C. bolteae* + *B. longum*) at a concentration of 1 × 10^5^ CFU/mL in 96 deep well plates containing 1.5 mL of YCFA and incubated anaerobically at 37 °C. Fructo-oligosaccharides (filter sterilized) were added to the media as the fermentable carbohydrate source to give a final concentration of 0.2% (*w*/*v*%). Cultures were sampled at selected time points. In addition, 200 uL aliquots (triplicates) of culture samples were pooled, frozen, and later used for metabolite quantification.

### 2.3. SCFA Quantification in Individual and Co-Culture Supernatants

Bacterial liquid cultures were grown for up to 24 h at 37 °C anaerobically. Cultures were sampled at specific time points, centrifuged at 16,000× *g* for 2 min, and the supernatants collected. Deuterium-labelled internal standards were added. Gas chromatography–mass spectrometry (GC–MS) was performed to detect acetic acid, propanoic acid, butyrate and lactate levels as previously described [7]. 

### 2.4. GVHD Murine Mouse Model

All animal experiments performed in this study were approved by the University of Texas Southwestern Medical Center’s institutional animal care and use committee. Mice were housed in a specific pathogen-free (SPF) facility under controlled environmental conditions, including a 12 h light/dark cycle, temperature of 20–22 °C, and humidity of 40–60%. They were provided with autoclaved bedding, standard chow, and autoclaved water *ad libitum*. Cages were changed regularly, and animals were monitored daily for signs of distress or illness. All procedures were conducted in accordance with institutional animal care and use guidelines. GVHD was induced in mice as previously described [5]. Briefly, female BALB/c mice (8–12 weeks of age; Jackson Laboratory, Bar Harbor, ME, USA) received oral levofloxacin (1 mg/mL) in their drinking water for seven days prior to stem cell transplantation (SCT). BALB/c mice underwent lethal irradiation followed by transplantation via intravenous injection of 2 × 10^7^ T cell-depleted bone marrow cells and 5 × 10^6^ splenic T cells from female C57BL/6 donors (8–12 weeks of age; Jackson Laboratory). Post-transplant, mice continued oral levofloxacin treatment until day +7, after which they received oral clindamycin from day +8 to day +10 to deplete commensal anaerobes. Beginning on day +11, mice were administered either an optimized and standard probiotic bacterial cocktail or sterile PBS via oral gavage every three days for the remainder of the study. The bacterial cocktail contained *Clostridium bolteae, Bifidobacterium longum, Blautia producta, Lactococcus lactis* (human, ATCC 19435, strain NCTC 6681), *Ruminococcus gnavus* (human, ATCC 29149, strain VPI C7-9), and *Ruminococcus torques* (human, ATCC 27756, strain VPI B2-51), with each strain administered at 1 × 10^8^ colony-forming units (CFU) in a total volume of 0.2 mL sterile PBS. Mice were monitored for up to 60 days for clinical GVHD scores, daily weight, and overall survival. Mice exhibiting severe weight loss (>20%) or significant morbidity were euthanized. Body weight and GVHD scores were analyzed using fitted slopes for individual mice and compared between groups using unpaired *t*-tests. Survival was monitored using Kaplan–Meier curves and analyzed with log-rank tests. Each experimental group consisted of five mice, with experiments conducted twice to achieve a total sample size of 10 per condition. To ensure a shared microbiota, littermates were co-housed within the same cages throughout the study.

### 2.5. Quantification of SCFA in Mouse Cecal Contents

100 mg of mouse cecal contents (fecal material) was transferred into a pre-weighed 1.5 mL microfuge tube containing 500 uL of sterile phosphate buffered saline (PBS). The microfuge tube was weighed and the exact weight of the cecal contents was then recorded. Microfuge tubes were vortexed at maximum speed for 2 min and then centrifuged at 6000× *g* at 4 °C for 15 min. The supernatants were collected and aliquoted. Deuterium-labelled internal SCFA controls were added to the tube. An aliquot of the reaction mixture was analyzed by gas chromatography–mass spectrometry (GC–MS) to detect acetic acid, propanoic acid, and butyric acid levels [7]. SCFA concentrations were determined using standard curves generated from the SCFA standards. 

### 2.6. qPCR for Microbiota Analysis

The relative abundance of the bacterial groups CLEPT (*Clostridium leptum* group; clostridial phylogenetic cluster IV, representing the *Ruminococcaceae* family) and ENTERO (*Enterobacteriaceae*) was quantified using quantitative PCR (qPCR) analysis with SsoAdvanced SYBR Green Supermix (Bio-Rad, Hercules, CA, USA). Group-specific 16S rRNA gene primers were used as previously described [8]. Bacterial abundance was determined by generating standard curves based on cloned DNA corresponding to a short segment of the 16S rRNA gene, which was amplified using conserved bacterial primers.

### 2.7. Statistical Analysis

Unless otherwise noted, GraphPad Prism v.9.2 was used for statistical analysis. All datasets were tested for normality (e.g., Shapiro–Wilk). Datasets with normal distribution were analyzed with parametric tests, such as the standard Student’s *t* test or one-way analysis of variance (ANOVA) with Bonferroni post-test. For nonnormal distributions, nonparametric tests, such as the Mann–Whitney *U* test or Kruskal–Wallis with Dunn’s post-test, were applied. Survival was analyzed using the Mantel–Cox log-rank test. A *p*-value < 0.05 was considered significant and is noted as follows: * *p* < 0.05, ** *p* < 0.01, *** *p* < 0.001, **** *p* < 0.0001. 

## 3. Results

### 3.1. Isolation of Gut Bacterial Strains

Using traditional culturing methods and the Prospector (Isolationbio) (Figure 1), we screened a total of 2602 bacterial colonies and were able to isolate several strains of three of our bacterial species of interest: *Clostridium bolteae* (two strains), *Blautia* spp. (six strains) and *Bifidobacterium longum* (six strains). 

### 3.2. Gut Microbiota (C. bolteae, Blautia spp., and B. longum) Strains Exhibit Significant Differences in Their Capacity to Produce Short-Chain Fatty Acids In Vitro

We sought to determine whether different strains of the same bacteria (*C. bolteae*, *Blautia* spp., and *B. longum*) differ in their capacity to produce short-chain fatty acids. “Standard” strains of these were obtained from American Type Culture Collection (ATCC). All bacteria were cultured in YCFA media. Culture supernatants were collected and analyzed by gas chromatography–mass spectrometry (GC–MS) to quantify SCFAs. Indeed, some newly isolated/cultured strains of *C. bolteae* produced significantly more butyrate and acetate when compared to the standard *C. bolteae* strain (designated as A), whereas propionate levels were not significantly different between the standard strain and the cultured strains (designated by numbers) (Figure 2A). In contrast, we observed no significant differences in butyrate production among the various *Blautia* strains, although some cultured strains produced more acetate and propionate than the standard strain (Figure 2B). Finally, one of the cultured *B. longum* strains produced more acetate, butyrate, and propionate compared to the standard strain (Figure 2C). In addition, this *B. longum* strain also was able to produce more lactate, which is notable as *Bifidobacterium* strains can produce lactate which is used by other gut microbiota taxa as a substrate [9]. Collectively, these results show that specific strains of the same bacteria can have varying functional capacities in producing short-chain fatty acids. The strains that produced the highest concentration of SCFAs (labeled “1”) were selected and used for the subsequent in vitro and in vivo experiments described below. 

### 3.3. B. longum Augments Butyrate Production by C. bolteae and Blautia spp. When Co-Cultured In Vitro

As noted above, some *Bifidobacterium* species are known to utilize non-digestible but fermentable dietary carbohydrates as growth substrates [9,10,11] to produce lactate and acetate. Butyrate-producing commensal bacteria that cannot utilize these dietary carbohydrates are able to utilize the lactate and acetate formed by *Bifidobacterium* spp., to produce more butyrate [9,12,13,14,15]. To evaluate whether *B. longum* could augment butyrate production by the other bacterial strains in vitro, we cultured the different butyrate-producing *C. bolteae* and *Blautia* species individually as monocultures and together with *B. longum.* Culture supernatants collected at 0, 4, 8, 12, and 24 h time points were analyzed by GC–MS to determine butyrate production. *Blautia* spp. (Figure 3A) and *C. bolteae* (Figure 3B) butyrate production was higher when co-cultured with *B. longum* as compared to monocultures. Strikingly, co-cultures of the isolated/cultured strains showed increased butyrate levels when compared to the standard strains of the same species (Figure 3C). These data demonstrate that some taxa (e.g., *B. longum)* can support the growth and production of specific metabolites (e.g., butyrate) of other gut microbiota taxa (e.g., *C. bolteae* and *Blautia* spp.).

### 3.4. An Optimized Precision Probiotic Therapy Enhances GVHD Mitigation, Prolongs Survival, and Increases SCFA Production in the Murine Gut

In our previous study, we found that supplementation with a probiotic consortium using “standard” strains of gut microbiota identified in our clinical study [5] improved GVHD and led to an enrichment of butyrate-producing gut microbiota in mice. We hypothesized that supplementation with optimized bacterial consortia that more efficiently produced SCFAs might be more effective at mitigating GVHD and increasing survival. Thus, utilizing a well-established murine model of allo-SCT and GVHD (Figure 4A), we treated transplanted mice with PBS (vehicle), standard probiotics (Pstd) using ATCC repository strains (as we did in our prior study [5]), or an optimized probiotic consortia (Popt) every 3 days for the remainder of the experiment. Weight and GVHD scores were recorded daily during the acute phase of GVHD. Administration of either the standard or optimized probiotics mitigated GVHD as evidenced by decreased weight loss (Figure 4B) and decreased GVHD clinical scores (Figure 4C), as compared to the group receiving vehicle. But, mice receiving the optimized probiotics exhibited greater GVHD mitigation and prolonged survival compared to those mice receiving the standard probiotic consortia (Figure 4B–D). 

To determine the effect of the probiotics on gut microbiota taxonomic composition, we first performed gut microbiota taxonomic profiling via 16s rRNA sequencing at various time points prior to and after stem cell transplant. Notably, 3 days after the onset of probiotic treatment, an expansion of Enterococcaceae and Enterobacteriaceae in the vehicle group and Akkermansiae in the treatment group was observed (Appendix A). Interestingly, optimized probiotics led to the decrease in *Akkermansia*, whose relative expansion in the colon of mice with GVHD has been shown to be associated with worse prognosis [16]. As 16S rRNA sequencing produces relative abundance data, we also performed bacterial qPCR over the duration of the experiment to assess absolute abundance of specific gut microbiota groups: CLEPT (*Clostridium leptum* group), a group of gut microbiota found to be protective against GVHD in our prior study [5] and also capable of producing SCFAs, and ENTERO (Enterobacteriaceae family), which includes members such as *E. coli*, *Shigella* spp., and *Klebsiella* spp. which have been shown to induce inflammation in the gut [17,18]. Indeed, CLEPT levels decreased while ENTERO levels increased after SCT (Appendix A). We found that GVHD-induced CLEPT depletion and ENTERO expansion (observed in the control group) was reversed with the administration of the standard and optimized probiotics.

Finally, to assess the functional impact of probiotic administration, we measured SCFA levels (via GC–MS) in cecal contents of mice treated with vehicle, standard probiotics, and optimized probiotics. Mice receiving the optimized probiotic had significantly increased levels of intraluminal cecal acetate and butyrate compared to mice treated with the standard probiotics (Figure 4E). These collective data suggest that probiotic therapeutic efficacy can be optimized by utilizing precision probiotics selection based on microbiota strain-level functional capacity to modulate host immune and/or metabolic pathways. 

## 4. Discussion

In this study, we sought to investigate the concept that “not all bacteria are created equal”, specifically, studying how different strains of the same gut bacterial species vary in their functional capacity to produce SCFAs, a gut microbiota-derived metabolite shown to be important for inducing immune tolerance and dampening inflammation. We posited that isolating specific gut microbiota strains identified as protective against GVHD in our prior clinical study from pediatric SCT patients who did not develop GHVD would allow us to culture and isolate “super” functional gut microbiota (in their capacity to produce SCFAs) and thereby produce an optimized precision probiotic therapy that would be more effective in mitigating GVHD than using standard strains available from a commercial repository. Our approach aligns with recent evidence supporting personalized microbial therapies over “one-size-fits-all” probiotic formulations [4,19].

The utility of using precision probiotics has been previously demonstrated in preclinical models of inflammatory bowel disease (IBD) and autism spectrum disorder (ASD) [20,21,22]. Biagoli et al. showed that using personalized microbiota resulted in decreased inflammation and enhanced gut barrier integrity, whereas commercial probiotic formulations had inconsistent effects [21]. Similarly, in autism spectrum disease models, precision microbiota from resilient hosts reduced behavioral symptoms more effectively than commercial probiotics, highlighting the importance of host-specific microbial compatibility [22,23]. These studies suggest that while commercial probiotics may offer generalized benefits, they often lack the specific strains or microbial characteristics necessary to influence host immune responses effectively in complex disorders. 

Different strains of the same bacterial species can exhibit variable phenotypes [24]. For example, different strains of the same species could differ in their ability to produce bacteria-derived metabolites that modulate host immune responses. Notably, the gut microbiota-derived metabolites, short-chain fatty acids (SCFAs), are critical for host immune-microbiota education and adaptation, epithelial homeostasis, and immune tolerance (as evidenced by SCFAs mitigating inflammation in inflammatory bowel disease preclinical models in the absence of the gut bacteria that produce them [25]). The SCFA butyrate has been shown to protect intestinal epithelial cells and decrease GVHD severity [26]. In our study, the increased efficacy of precision probiotics is likely attributed to targeted immune modulation via gut microbiota-derived SCFAs. SCFAs directly modulate intestinal epithelial cell damage [26] and are key regulators of regulatory T cell (Treg) homeostasis, enhancing Treg proliferation and function [25]. In inflammatory bowel diseases, such as ulcerative colitis and Crohn’s disease, decreased SCFA production has been associated with exacerbated gut inflammation and dysbiosis [27]. Furthermore, SCFAs influence the epigenetic regulation of immune cells, as butyrate acts as a histone deacetylase (HDAC) inhibitor, modulating gene expression to favor anti-inflammatory pathways [28]. Notably, SCFAs alone (in the absence of the gut microbiota that produce them) regulate colonic T cell homeostasis and can mitigate inflammation in preclinical colitis [25] and GVHD models [26]. 

Bacterial strains of the same species can also exhibit substantial functional differences beyond SCFA production, impacting host health and disease outcomes. For example, distinct strains of *Bacteroides fragilis* have been shown to produce polysaccharide A, a molecule that promotes regulatory T cell (Treg) induction and immune tolerance [29]. *Bifidobacterium longum* is selected for human milk oligosaccharide utilization in breastfeeding infants [30], whereas closely related *B. longum* strains in the adult gut frequently possess the capacity to ferment carbohydrates, but not human milk oligosaccharide [31]. Studies on *Lactobacillus reuteri* have shown strain level differences in bile salt hydrolase (BSH) activity. High-BSH activity *L. reuteri* strains contribute to host cholesterol reduction and bile acid modification, unlike their low-BSH counterparts, resulting in differential impacts on gut health and metabolic pathways [32]. Ultimately, variations among microbial strains influence not just the biology of the host and the functionality of individual microbes, but also the structure and evolutionary relationships within the broader gut ecosystem. 

The precision probiotics approach holds promise for a variety of immune-mediated conditions, such as inflammatory bowel disease, cancer immunotherapy, and autoimmune diseases, where microbial imbalances and inflammatory responses play key roles in their pathogenesis [33]. Translating these promising preclinical results to clinical practice, however, involves several critical steps including regulatory oversight, dose optimization studies, quality control, and the development of precise manufacturing protocols to ensure reproducibility and safety across patient populations. As the field of precision medicine advances, overcoming regulatory and logistical barriers in the clinical application of live microbial therapeutics will be critical to realizing the potential of microbiome-based interventions for GVHD and related immune-mediated diseases. Ultimately, our findings provide a foundation for the next steps in clinical translation, which include dose-finding studies, patient stratification criteria, and robust clinical trial design to validate the efficacy and safety of precision probiotics.

A limitation of this study is our reliance on a mouse model, though valuable for preclinical insights, it does not fully replicate the complexity of human GVHD. Additionally, individual differences in microbial communities may lead to variable strain efficacy across patients, complicating the standardization of precision probiotics. Finally, our study’s relatively short-term follow-up in mice limits understanding of the long-term impacts of precision probiotics. While initial results indicate improved survival and reduced GVHD severity, these benefits may evolve or diminish over time, highlighting the need for extended studies to assess sustainability and potential delayed effects of treatment. Addressing these limitations will be essential for advancing this approach toward clinical applications. 

In summary, by capitalizing on unique microbial adaptations found in disease resistant hosts, our study builds on the idea that harnessing disease resistant host-specific microbes can yield superior outcomes in disease modulation compared to non-specific “standard” strains, particularly in cases where immune tolerance is crucial. Exploring the specific molecular pathways modulated by precision probiotics could deepen our understanding of how these strains influence immune regulation and disease severity. Detailed pathway analysis may reveal novel immune or metabolic targets that could be leveraged to refine therapeutic strategies. Long-term studies are also essential to evaluate the stability and safety of precision probiotics over time, particularly across diverse populations who may have varied microbiome compositions and immune profiles. Collaborative studies will be crucial to addressing these gaps, including metabolomic analyses to identify microbial metabolites that play a protective role in diseases, which may illuminate new biomarkers or therapeutic candidates.

## Figures and Tables

**Figure 1 microorganisms-13-00706-f001:**
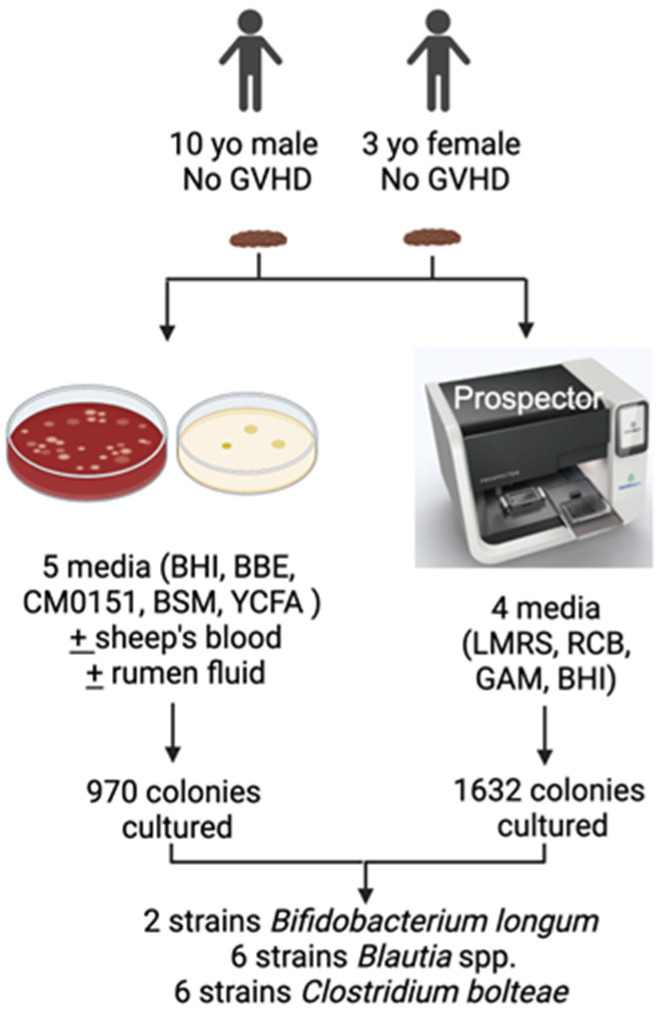
Stool from stem cell transplant (SCT) patients without graft-versus-host disease (GVHD) yielded several strains of bacteria shown to be protective against GVHD. Several culture media and the Prospector were used to inoculate stool suspension. Individual bacterial colonies were selected, re-streaked, and identified. Culturomics yielded 14 strains of three gut microbiota of interest. All bacteria were cultured at 37 °C in an anaerobic chamber. BHI, brain heart infusion. BBE, Bacteroides Bile Esculin. CM0151, Reinforced Clostridia medium. BSM, Bifidus Selective Medium. YCFA, Yeast extract, Casitone and Fatty acid. LMRS, Lactobacillus de Man–Rogosa–Sharpe medium. RCB, Reinforced Clostridial broth. GAM, Gifu Anaerobic Medium.

**Figure 2 microorganisms-13-00706-f002:**
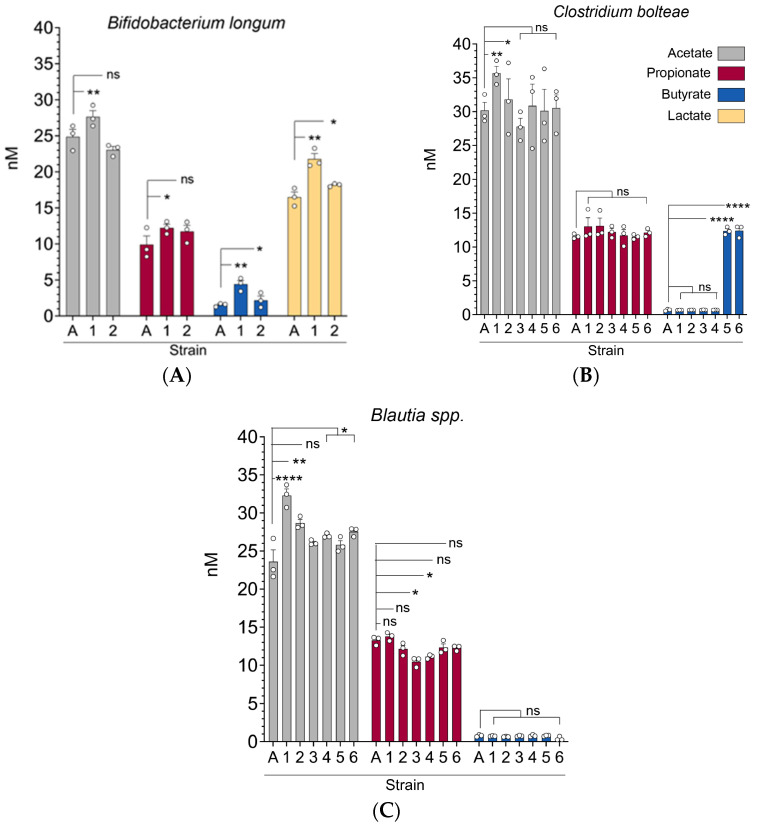
Significant strain-to-strain variability in short-chain fatty acid production for *B. longum* (**A**), *C. bolteae* (**B**), and *Blautia producta* (**C**). Commercially obtained strains (designated A in bar graphs) compared with strains cultured from stool of SCT patients without GVHD (numbered). Microbiota were grown in Yeast extract, Casitone and Fatty acid (YCFA) media. At 24 h, culture supernatants were collected and SCFAs quantified using gas chromatography with tandem mass spectrometry (GC–MS). Bars represent the mean ± SEM, representing three independent experiments, each with three technical replicates. Statistical analysis by Mann–Whitney test. * *p* < 0.05. ** *p* < 0.01. **** *p* < 0.0001. ns, not significant.

**Figure 3 microorganisms-13-00706-f003:**
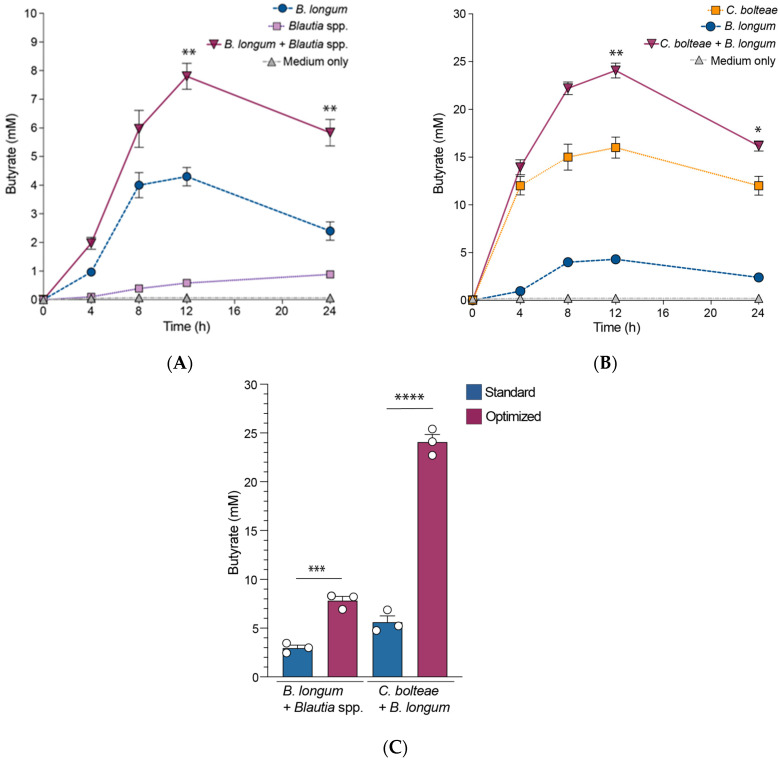
*Bifidobacterium longum* augments butyrate production by *C. bolteae* and *Blautia producta* in vitro. *Blautia* spp. (**A**) and *C. bolteae* (**B**) were cultured alone and with *B. longum*. Culture supernatants were collected every 4 h up to 12 h and then at 24 h, and butyrate was quantified using gas chromatography with tandem mass spectrometry (GC–MS). No butyrate was detected in uninoculated media. Data are mean ± SEM of three independent experiments; triplicate culture wells were pooled for butyrate analysis. (**C**) Comparing butyrate production in co-cultures of *C. bolteae*, *Blautia producta*, and *B. longum* in the standard and optimized probiotics cocktails. Statistical analysis by Mann–Whitney test. * *p* < 0.05, ** *p* < 0.01, *** *p* < 0.001 and **** *p* < 0.0001.

**Figure 4 microorganisms-13-00706-f004:**
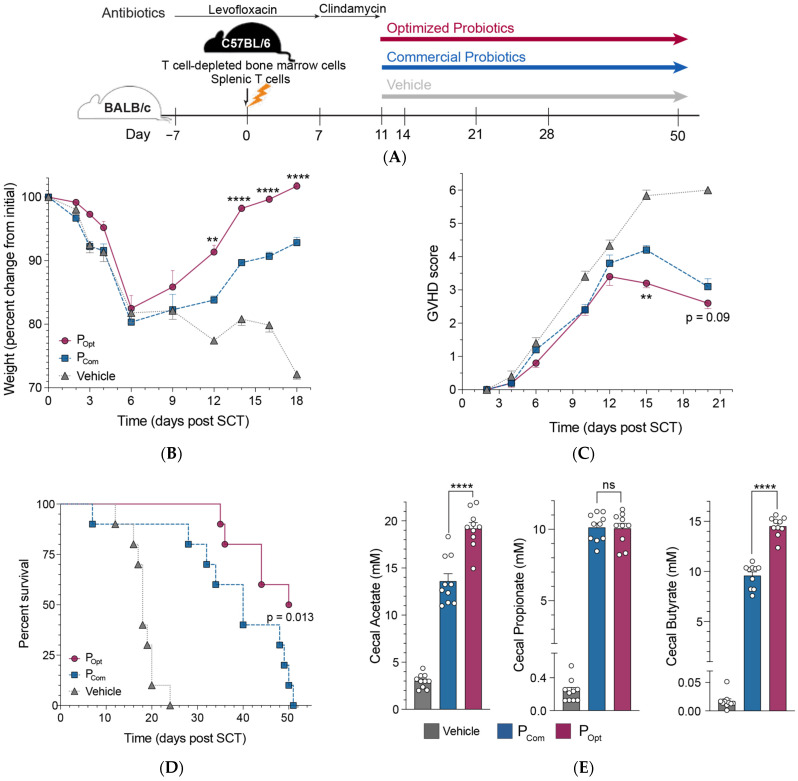
Oral supplementation with precision probiotics ameliorates GVHD and increases survival. (**A**) Schematic overview of the GVHD protocol used to assess probiotic efficacy in mice. Lethally irradiated Balb/C recipients underwent transplantation with 2 × 10^7^ C57BL/6 T cell-depleted bone marrow cells and 5 × 10^6^ splenic T cells. Recipients were initially treated with levofloxacin. After transplantation (day 0), mice were treated with levofloxacin in the drinking water until day +7, then clindamycin until day +10. Starting on day +11 mice were orally gavaged a probiotic cocktail (1 × 10^8^ colony-forming unit of *Clostridium bolteae*, *Ruminococcus gnavus*, *R. torques*, *Blautia producta, Bifidobacterium longum and Lactococcus lactis* in 0.2 mL sterile PBS) or sterile PBS (vehicle, 0.2 mL) every 3 days for the remainder of the experiment. (**B**) Weight and (**C**) GVHD scores were monitored during the acute phase of GVHD. (**D**) Survival of GVHD mice. Statistical analysis by log-rank test. (**E**) Acetate, propionate, and butyrate concentrations in cecal contents of GVHD mice. Bars represent mean ± SEM. Points represent results from individual animals. Data are 5 mice per group from 2 experiments, final *n* = 10 per group. Statistical analysis by Mann–Whitney test. ** *p* < 0.01. **** *p* < 0.0001. ns, not significant.

## Data Availability

The data that support the findings of this study are openly available at 10.6084/m9.figshare.28327319.

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
