# Peer review of "Optimizing Precision Probiotics for Mitigating Graft-Versus-Host Disease"

_microorganisms, 2025, doi:10.3390/microorganisms13040706_

Round 1
Reviewer 1 Report
Comments and Suggestions for Authors
The authors present the results of the studies performed on lab animals regarding the benefits of commensal bacterial species in lab animals with cell transplants and graft-versus-host disease (GVHD). The manuscript is well written, the methodologies used are appropriate, and the results obtained are presented concisely and logically. The authors used both microorganisms isolated from the specific mouse's microbiota and standardised microorganisms from a recognised international collection (ATCC) in their experiments to quantify microorganism metabolites (short-chain fatty acids) by GC-MS and assess their potential role.
The results obtained at this stage suggest that the precise delivery of specific probiotics can lead to increased therapeutic efficacy for GVHD. This finding represents the novelty of this article, and for this reason, I recommend its publication.
However, minor modifications need to be made to the manuscript before publication, as follows:
1)A sentence cannot begin with an abbreviation. The authors must carefully review the manuscript and rewrite all such sentences;
2) Before the chapter entitled Methodology, authors must write more clearly, in (3-5) sentences, the main objectives of the studies presented in their manuscript;
3)The scientific names of microorganisms must be written in italics.
4)The references must be formatted according to MDPI guidelines.
Reviewer 2 Report
Comments and Suggestions for Authors
Optimizing precision probiotics for mitigating graft-versus host disease
Abstract: add the objectives, experimental design, P value.
L20-23: Too long sentence. Here, we sought to optimize a six species precision probiotic therapy for graft-versus host disease (GVHD), an autoimmune complication following allogeneic stem cell transplantation, that we previously identified as associated with protection against GVHD in a prior clinical study and validated in a preclinical GVHD model.
You need to add a paragraph in the introduction about graft-versus host disease (GVHD).
L70: We utilized two different methods to isolate our gut bacteria of interest from fecal specimens. Revise, using we and our gut bacteria is misleading. Here and everywhere else
In vivo work: A brief description of the housing and environment that the mice lived in will be appreciated. No mentation about the experimental design and the procedure followed with the animals during the experiment.
L172-188: you need to focus on the results you obtained. This section provides general information with little results.
L240: what was the dose of fructo-oligosaccharide?
